# 3D Hierarchical Nanocrystalline CuS Cathode for Lithium Batteries

**DOI:** 10.3390/ma14071615

**Published:** 2021-03-26

**Authors:** Gulnur Kalimuldina, Arailym Nurpeissova, Assyl Adylkhanova, Nurbolat Issatayev, Desmond Adair, Zhumabay Bakenov

**Affiliations:** 1Department of Mechanical and Aerospace Engineering, School of Engineering and Digital Sciences, Nazarbayev University, Kabanbay Batyr Ave. 53, Nur-Sultan 010000, Kazakhstan; dadair@nu.edu.kz; 2National Laboratory Astana, Nazarbayev University, Kabanbay Batyr Ave. 53, Nur-Sultan 010000, Kazakhstan; arailym.nurpeissova@nu.edu.kz (A.N.); nurbolat.issatayev@nu.edu.kz (N.I.); zbakenov@nu.edu.kz (Z.B.); 3Department of Chemical and Materials Engineering, School of Engineering and Digital Sciences, Nazarbayev University, Kabanbay Batyr Ave. 53, Nur-Sultan 010000, Kazakhstan; assyl.adylkhanova@nu.edu.kz

**Keywords:** copper sulfide, lithium-ion battery, Cu foam, cathode, binder-free

## Abstract

Conductive and flexible CuS films with unique hierarchical nanocrystalline branches directly grown on three-dimensional (3D) porous Cu foam were fabricated using an easy and facile solution processing method without a binder and conductive agent for the first time. The synthesis procedure is quick and does not require complex routes. The structure and morphology of the as-deposited CuS/Cu films were characterized by X-ray diffraction and scanning electron microscopy coupled with energy-dispersive X-ray spectroscopy and transmission electron spectroscopy, respectively. Pure crystalline hexagonal structured CuS without impurities were obtained for the most saturated S solution. Electrochemical testing of CuS/Cu foam electrodes showed a reasonable capacity of 450 mAh·g^−1^ at 0.1 C and excellent cyclability, which might be attributed to the unique 3D structure of the current collector and hierarchical nanocrystalline branches that provide fast diffusion and a large surface area.

## 1. Introduction

Nowadays, lithium-ion batteries (LIBs) have become an indispensable part of many technology advancements by powering a variety of devices from miniature medical applications to large stationary grid energy storage. To continue to power sophisticated innovative technologies, LIBs need to be upgraded both in energy and power characteristics and in structural design (micro, flexible). Usually, the anodes in a battery system have sufficient capacity, unlike cathodes, which need greater attention to enhance the specific energy of LIBs. A variety of different cathodes have been researched, with some of them being commercialized. However, the search for a better cathode material equal to the conventional anode is still ongoing. 

Recently, sulfur and sulfur-containing compounds have attracted great attention due to their high theoretical capacity and abundance in nature [1]. Metal sulfides (Cu_x_S_y_, Ni_x_S_y_, etc.), in particular, have been reported as showing impressive electrochemical properties such as long cyclability and high rate capability [2,3,4,5,6]. Among these materials, copper sulfides are particularly attractive due to their high metal-like conductivity (1 × 10^−3^ S·cm^−1^), relatively large availability, low cost, and environmental-friendliness. Copper sulfides (Cu_x_S, x = 1–2) are binary and exist in variety of nonstoichiometric forms that vary from copper-rich Cu_2_S to copper-deficient CuS_2_, including intermediate Cu_1.96_S, Cu_1.8_S, Cu_1.75_S, Cu_1.6_S, Cu_1.39_S, and CuS with the moderate theoretical specific capacities varying from 337 to 560 mAh·g^−1^ [7]. Though many Cu_x_S cathodes have been investigated, the Li/Cu_x_S system still cannot be adopted for commercial purposes due to poor cycle life, low utilization of the active mass, and low Coulombic efficiency, which is linked to lithium polysulfide dissolution during the discharge reaction [8,9]. Additionally, little attention has been paid to the volumetric variation of Cu_x_S during the lithiation process, which leads to pulverization of the active material and the reduction in electrochemical cycle life [10]. At the same time, it is worth mentioning that the synthesis procedures for Cu_x_S compounds are cumbersome, requiring complex procedures, high temperatures, and the use of additional instruments, additives, and catalysts, which will increase the cost of the preparation process.

One of the most promising routes to mitigate the degradation of many electrodes lies in designing architectures. Mainly three-dimensional (3D) structured electrodes are evolving as smart electrodes with unique features such as interconnected porous networks and intrinsic structural integrity, avoiding the use of conductive agents and binders as well as continuous paths within the electrodes, facilitating fast transport of both lithium ions and electrons [11,12]. Bases on this, Adylkhanova et al. previously reported on the possibility of the Cu_x_S/Cu (x = 1.96, 1.8) foam structure, which showed reasonable performance [13]. It was also reported by Kalimuldina et al. that excess Cu cations at the surface of Cu foil, introduced to Cu_x_S, lead to the formation of Cu_1.96_S instead of CuS in the fully charged reaction, and this enhances the efficient utilization of Li_2_S_x_, leading to excellent capacity retention [14,15]. 

Taking into account all the above considered improvement routes, here, we have employed 3D porous Cu foam as a current collector to deposit thin CuS compounds. Cu foam has been known as an excellent flexible current collector due to its intrinsic properties, such as superior electrical conductivity, which is beneficial for electron transport. In addition, 3D scaffolding can maintain more porous architecture and active materials that could supply efficient electrolyte diffusion channels enabling the excellent electrochemical performance of an electrode. The excess amount of Cu in Cu foam will improve the cyclability, utilizing Li_2_S_x_. Hence, in this study, we have developed an innovative solution processing method to prepare 3D conductive CuS nanocomposites for the first time. 

## 2. Materials and Methods

### 2.1. Preparation of 3D Structure Cu_x_S

The Cu_x_S (1 < x < 1.8) thin films were directly grown on commercially available Cu foam (purity > 99.99%, porosity: 70~80%, 0.17–0.22 mm thickness, MTI Corp., Richmond, CA, USA). The detailed synthesis procedure starts with the heating of dimethyl sulfoxide (DMSO, Sigma Aldrich, St. Louis, MO, USA) from 80 to 115 °C and dissolving of S powder under vigorous stirring for 30 min to obtain a homogenous solution. The constant 0.1 g of S was dissolved at different temperatures to study the effect of temperature on the reaction. Then, the amount of solved S varied from 0.1 to 0.4 g per 20 mL solution at the fixed temperature of 115 °C. After fully dissolving S in DMSO, the Cu foams that were pre-washed with acetone and dried were soaked in that solution for 3 min. Within 3 min, the Cu foam was fully covered with a black-colored precipitate of Cu_x_S. The Cu_x_S that was obtained on Cu foam was then dried in the oven at 60 °C for 24 h. 

### 2.2. Characterization of Electrode Materials

The obtained Cu_x_S thin films on Cu foam were characterized by X-ray powder diffraction (Rigaku SmartLab^®^ X-ray diffraction (XRD) system, Tokyo, Japan) to investigate the phases of Cu_x_S. Scanning electron microscopy (SEM) using a Crossbeam 540 coupled with energy dispersive X-ray spectroscopy (EDS) was employed to observe the morphologies of the thin films. TEM was employed to observe the grain sizes and morphology of Cu_x_S. Elemental compositions of CuS nanocomposite were investigated using X-ray photoelectron spectroscopy (XPS, NEXSA Thermo Scientific, Waltham, MA, USA). 

### 2.3. Cell Fabrication and Electrochemical Characterizations 

To test the electrochemical performance of obtained cathodes, the as-prepared binder-free Cu_x_S/Cu foams were cut into 16 mm disks. Coin-type cells of CR2032 were assembled in an argon-filled glove box using Li foil as a reference electrode and 1 M lithium bis(trifluoromethane sulfonamide) (LiTFSI) in 1,3-dioxolane (DOL) and dimethoxyethane (DME) with a volume ratio of 1:1 as an electrolyte. The mass loading of CuS active material varied between 8.7 and 10.4 mg·cm^–2^. Celgard 2400 microporous polypropylene was used as the separator membrane. The cells were tested in the voltage range between 1.0 and 3.0 V with a multichannel battery test system Arbin, which had a constant current in the form of a 0.1 C rate (1 C = 7–7.3 mAh). Electrochemical impedance spectroscopy (EIS) measurements were performed on a CHI660C electrochemical workstation under open-circuit conditions over a frequency range of 0.01 Hz to 100 kHz by applying an AC signal of 5 mV in amplitude throughout the tests.

## 3. Results and Discussion

The Cu_x_S phase was studied by XRD analysis of the as-prepared samples. Figure 1 demonstrates the spectra of CuS, Cu, and Cu_1.8_S at different dissolved S content in DMSO. Since the conditions for temperature and time were fixed, the differences in the XRD peaks were associated only with the influence of the S content. The two high-intensity peaks at 43.3 and 50.43° are typical of Cu foam (ICDD: 004-0836). In the samples with 0.1 to 0.3 g of S content in the DMSO solution, a mix of the spectra of CuS is observed (ICDD: 006-0464) and Cu_1.8_S (ICDD: 01-073-8624) (Figure 1b) [8]. All the peaks are mainly indexed to CuS, though there is a clear additional peak of Cu_1.8_S at 46.10°. 

Further, when the S content was increased to 0.4 g in the solution, all the peaks correspond to hexagonal CuS. It is worth mentioning the reaction will only take place at a stabilized temperature of 115 °C as the heat facilitates the reaction time between Cu and S. DMSO is a solvent with a high boiling point of 189 °C, and it shows the solubility of sulfur at relatively high temperatures above 115 °C. Therefore, DMSO can be used as an efficient solvent for the preparation of S nanocomposites such as metal sulfides. Once S has been fully dissolved in the heated DMSO solution, Cu foam was added to the solution then removed after 3 min. Through the manipulation of the S content, we could obtain the desired well-crystallized CuS phase with a space group of P6_3_/*mmc* at 115 °C and 3 min reaction time with strong and sharp peaks [16]. 

In order to confirm the formation of pure CuS, XPS analysis was conducted. The XPS technique was used to understand the structure of CuS because the chemical or physical environment may affect the valence states of the films. In Figure 2, XPS analysis confirmed the formation of CuS as we can observe Cu 2p and S 2p peaks. High-resolution XPS peaks demonstrated peaks of Cu 2p_3/2_ at 932.35 eV and Cu 2p_1/2_ at 952.20 eV (Figure 2a) and corresponding peaks of S 2p_3/2_ at 161.5 eV and S 2p_1/2_ at 162.4 eV (Figure 2b), respectively. All the defined Cu and S peaks are consistent with CuS peaks reported in the literature [17,18]. The weak satellite shake-up peaks detected at 943 eV define the existence of the Cu^2+^ state only in the sample (Figure 2a). 

To expose the morphology and microstructure of interconnected CuS nanosheet arrays, SEM analysis was performed. Cu foam can easily be converted to Cu_x_S with the addition of S. Therefore, the obtained Cu_x_S nanowall arrays were clearly and closely attached on the surface of Cu foam due to the simple reaction activity between Cu foam and S. Figure 3a,b demonstrates an images of the sample with the lowest 0.1 g of S content in the DMSO solution and the formation of small nanosheets can be seen. The increase of S content to 0.4 g allows the growth of intense Cu_x_S nanosheets on the Cu foam with clear morphology evolution (Figure 3e). The top-view SEM images show that the interconnected CuS nanosheets are uniformly allocated and affix firmly to the surface of the Cu foam with an average thickness of around 500 nm (Figure 3f). The magnified SEM image in Figure 3f shows that the interconnected array of CuS nanosheets have coarse surfaces which in turn consist of myriad ultrathin nanosheets developed on both sides. The as-formed unique interconnected arrays of nanosheets with abundant open surface areas and pores can contribute to more electroactive surface sites, which could result in effective penetration of the electrolyte and enhancement of mass/charge transfer at the electrode/electrolyte interface.

The elemental distribution of CuS was further investigated by EDS analysis with the results shown in Figure 4a–c as the map of chemical elements of Cu and S. The images demonstrate that both elements are well distributed, with particular attention paid to the S distribution within the Cu foam framework. The well-distributed S elements show that the reaction to form CuS throughout the surface of Cu foam was uniform. The HRTEM image in Figure 5 exhibits the lattice fringes displaying the interplanar spacing of 0.36 nm being in agreement with a hexagonal CuS [19]. 

A schematic diagram that shows the growth of the nanosheet arrays with three-dimensional hierarchical branched structured CuS on the Cu foam surface is shown in Figure 6a. As was discussed, S in the DMSO solution is dispersed evenly and, moreover, a sufficient amount of Cu exists in the interiors of the Cu foam framework. The free Cu^2+^ derived from the Cu foam would proceed to react instantly with the S in solution, while the secondary reaction takes place along the surface of the first nanosheets nearest to the Cu foam; hence, the principle of proximity. Furthermore, the freshly formed arrays of nanosheets increase in size with the increasing S content in the solution, considering the active sites present as analogous to the first nanosheets. In Figure 6b, the flexibility test of CuS on Cu foam has been demonstrated. The electrode shows a bendable structure without breakage. 

The charge–discharge profiles of CuS film on Cu foam are presented in Figure 7a. The first charge and discharge capacity reaches 420 mAh·g^−1^ with excellent reversibility. We can see the typical CuS first discharge plateaus at the 2.1 and 1.7 V regions. The first charge shows a flat plateau at 2.25 V and decreases and flattens after 10 cycles to 2.1 V. This phenomenon has been previously demonstrated by several works and was associated with the phase transformation of hexagonal CuS to the copper-excess phase of tetragonal Cu_1.96_S. At the 10th cycle, charge–discharge capacity reaches 430 mAh·g^−1^. The increase in capacity could be related to the efficient active sites for the reversible reaction between Cu and Li_2_S during charging as the system has a sufficient amount of Cu for the reaction. 

Cycling performance at 0.1 C is presented in Figure 7b, and it can be observed that there is stable cycling at 430 mAh·g^−1^ for 40 cycles, which is relatively higher that other thin films of CuS reported in the literature. For example, electrodeposited 50 nm CuS thin film, investigated by Chen et al. [20], delivered initial discharge capacity equal to 545 mA h g^−1^ at a rate of 0.05 C. When the electrode was subjected to prolonged cycling, the capacity dropped drastically to below 100 mAh·g^−1^ due to the solubility of lithium polysulfides. Another CuS obtained by in situ preparation method reported by Wang and co-workers exhibited 400 mAh·g^−1^ over 30 cycles [21]. In our work, the mass loading of CuS was high due to the penetration and complete internal formation of CuS on the entire porous structure of the 3D Cu foam. Coulombic efficiency is about 100% for 40 cycles, exhibiting excellent reversibility of the generally perceived (CuS + 2Li^+^ → Li_2_S + Cu + 2e^−^) reaction activity. However, previous studies state that the CuS reaction with Li^+^ takes more complex steps, where during the first discharge, new copper-excess Cu_1.96_S phases are formed. It is worth mentioning that the reversible charge reaction between Cu and Li_2_S forms low-crystalline CuS, leading to capacity deterioration. However, the presence of Cu foam with the additional Cu cations facilitates better reaction activity and reversible formation of CuS [14,15,22]. 

The discharge reactions occurring at the first plateau at 2.1 V were defined through in situ XRD analysis as follows [23]:CuS + *x*Li^+^ + *x*e → Li_x_CuS(1)
Li_x_CuS + (1 *− x*)Li^+^ + (1 *− x*)e^−^ → 0.5Cu_1.96_S + 0.5Li_2_S + 0.02Cu(2)

The second plateau, defined at 1.7 V, has been defined through the following reaction:0.5Cu_1.96_S + Li^+^ + e^−^ → 0.5Li_2_S + 0.98Cu(3)

On the other hand, the general charge reaction was shown through the following equation: Li_2_S + Cu → CuS + 2Li^+^ + 2e^−^(4)

The capacity dependence of CuS on the current density is shown in Figure 8a. The 3D hierarchical nanocrystalline CuS cathode was able to show stable capacity retention at increased current rates, showing ~80% capacity retention at as high as the 2 C rate. This can be explained by the superior conductivity of CuS. Electrochemical impedance measurements provided further evidence of the conductivity of CuS. The plot depicted in Figure 8b includes a semicircle that appears at the high-frequency range and a straight line (Warburg tail) at the low-frequency range. The semicircle accounts for a charge-transfer resistance, while the 45° line represents the diffusion-resistance of Li^+^ ions in the bulk structure of CuS. The fitted parameters for the equivalent circuit of impedance spectra were R1 = 2.58 ohm, R2 = 1.236 ohm, Q2 = 12.65e^−6^ F, R3 = 1.549 ohm, and Q3 = 0.074 F. From Figure 8b, small charge-transfer resistance of around 1.2 ohm can be observed, which is due to the unique structure of CuS allowing an easy charge-transfer. The post-mortem SEM image of the cycled electrode after 5 cycles is shown in Figure 8c. We can observe the nanocrystalline structure of CuS with slight agglomeration. The structure still preserves the easy access of the electrolyte and Li-ions to maintain the stable cycling performance. 

## 4. Conclusions

The hexagonal crystalline structure CuS with a space group of P6_3_/*mmc* was successfully prepared by a straightforward solution method using S dissolved in DMSO solution and Cu foam. The 3D architectural and nanocrystalline branched CuS electrode demonstrated excellent electrochemical performance with a reversible capacity of 420 mAh g^−1^ at a 0.1 C rate. Moreover, the rate capability at 2 C exhibited a capacity of 400 mAh g^−1^, attributed to the unique structure of CuS foam that effectively increased the contact area between electrolyte and electrode. Therefore, the binder-free highly conductive CuS electrode could be a potential cathode for high-performance rechargeable battery applications.

## Figures and Tables

**Figure 1 materials-14-01615-f001:**
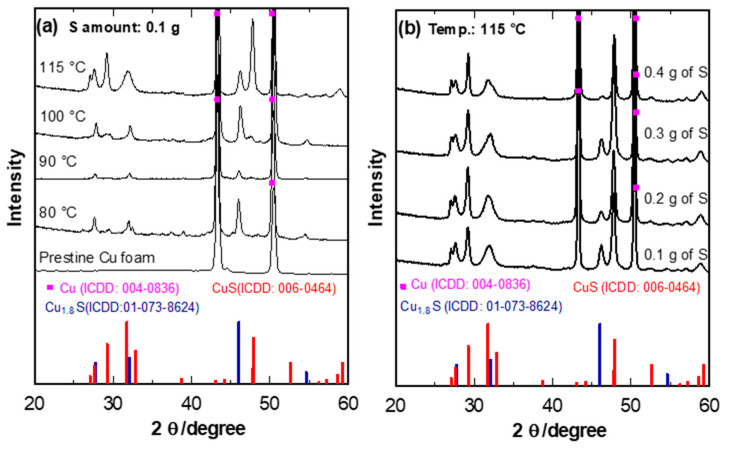
XRD patterns of Cu_x_S on Cu foam (**a**) at different synthesis temperatures with S amount of 0.1 g dissolved in DMSO; and (**b**) samples at 115 °C for 3 min with different amounts of S dissolved in DMSO.

**Figure 2 materials-14-01615-f002:**
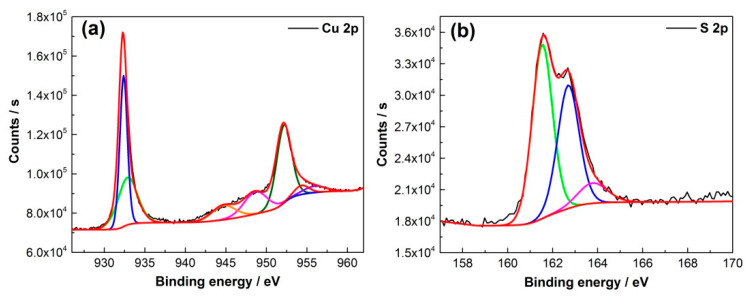
XPS patterns of CuS on Cu foam: (**a**) Cu2p and (**b**)S2p.

**Figure 3 materials-14-01615-f003:**
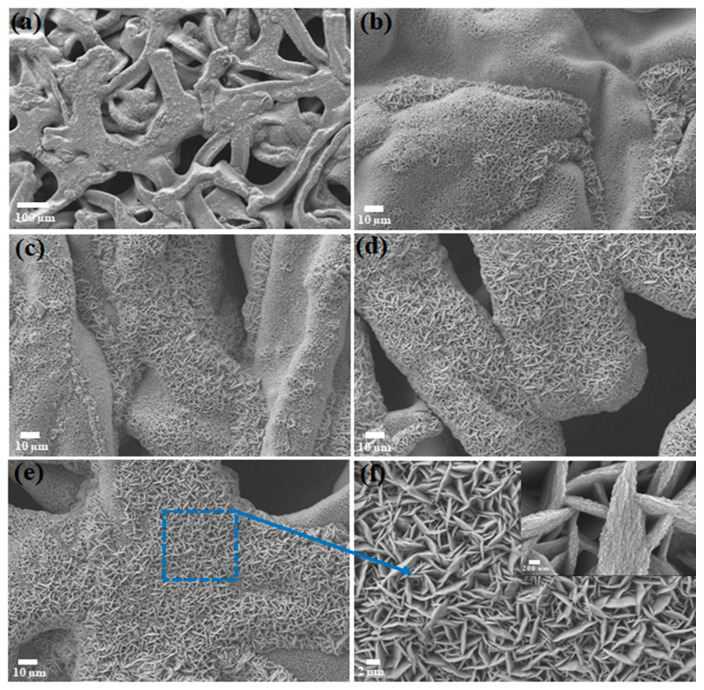
SEM images of Cu_x_S with different sulfur contents: (**a**) Cu_x_S-coated Cu foam; (**b**) 0.1 g of S; (**c**) 0.2 g of S; (**d**) 0.3 g of S; (**e**,**f**) 0.4 g of S with different magnifications.

**Figure 4 materials-14-01615-f004:**
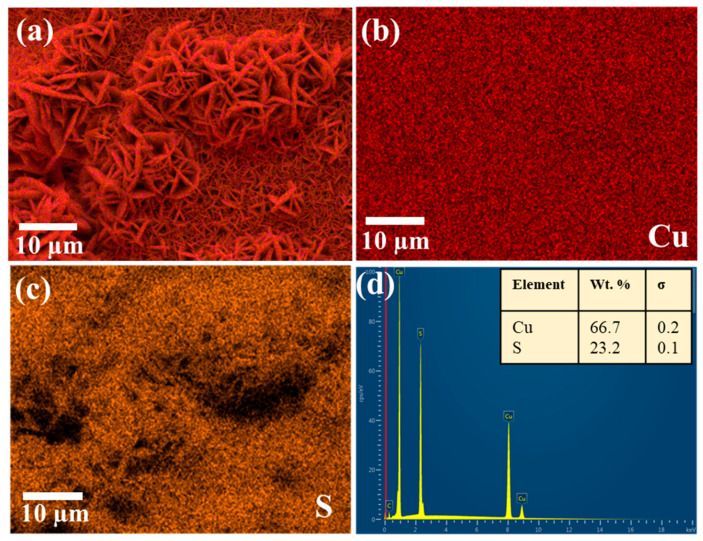
EDS mapping of Cu and S elements on the (**a**–**c**) surface of CuS nanocrystalline branches. (**d**) EDS spectra.

**Figure 5 materials-14-01615-f005:**
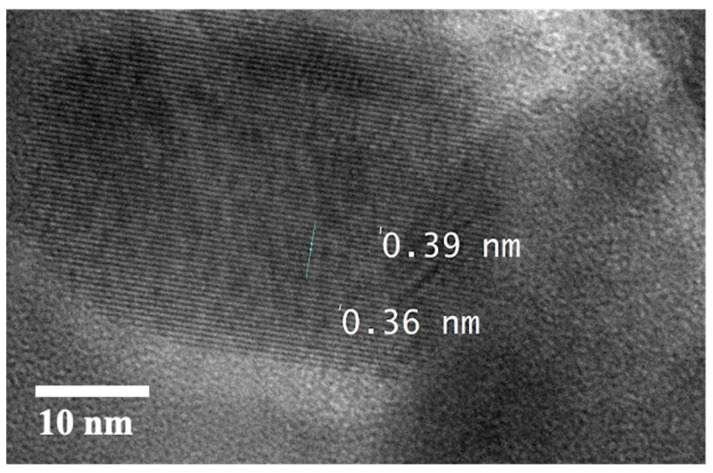
TEM image of a hexagonal CuS.

**Figure 6 materials-14-01615-f006:**
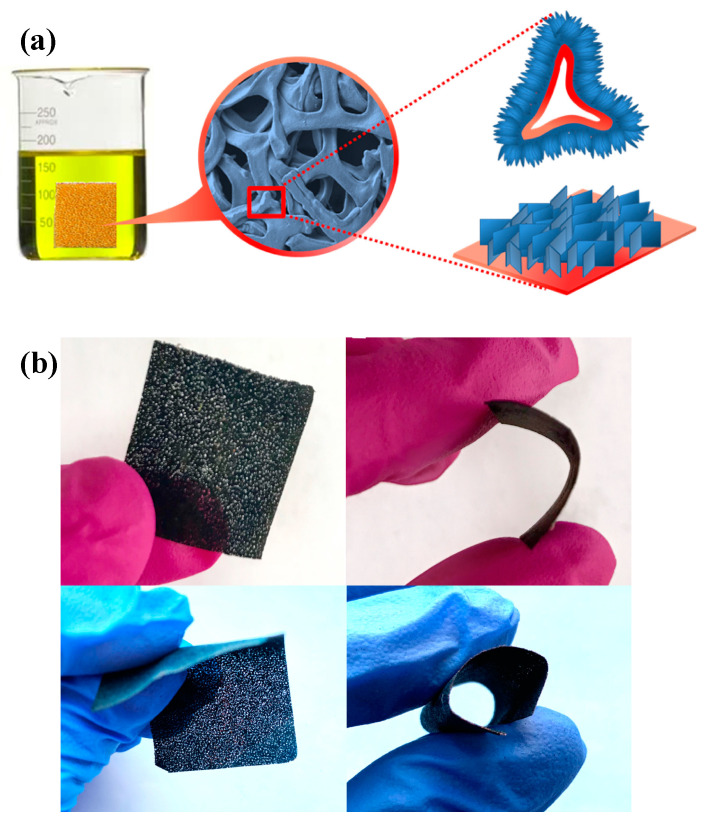
(**a**) Schematic diagram of the CuS formation process on the surface of Cu foam. (**b**) Flexibility test of CuS formed on Cu foam.

**Figure 7 materials-14-01615-f007:**
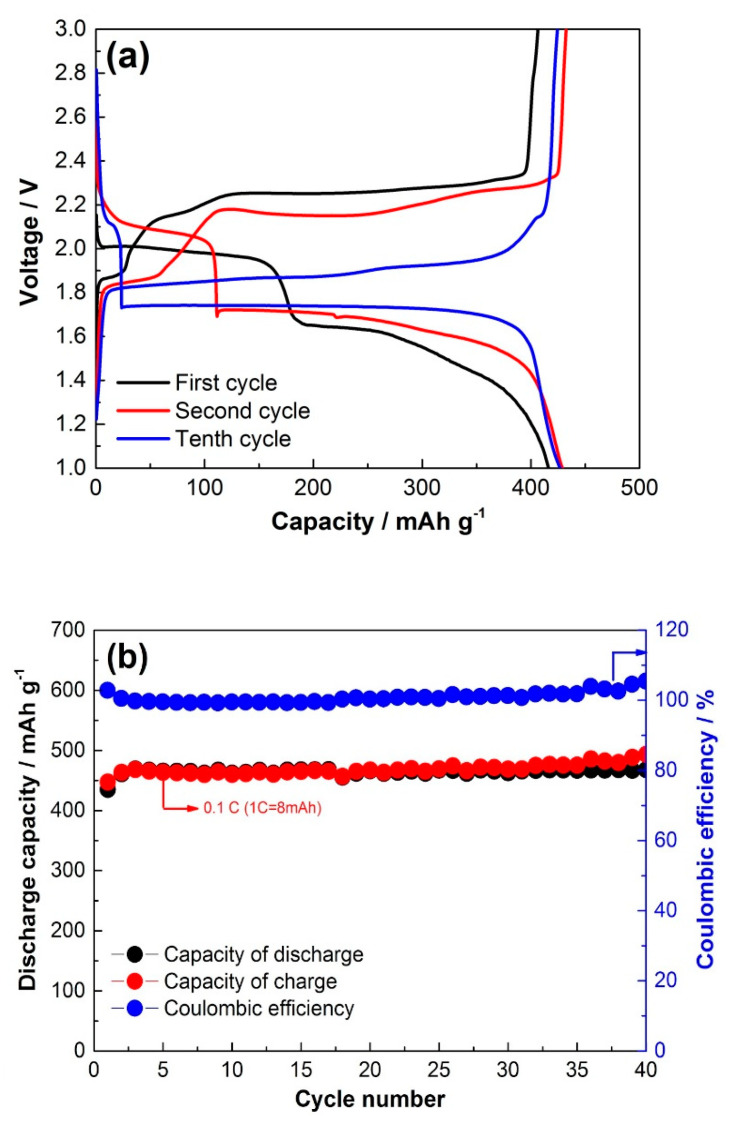
(**a**) Electrochemical charge–discharge and (**b**) cycling performance of CuS on Cu foam.

**Figure 8 materials-14-01615-f008:**
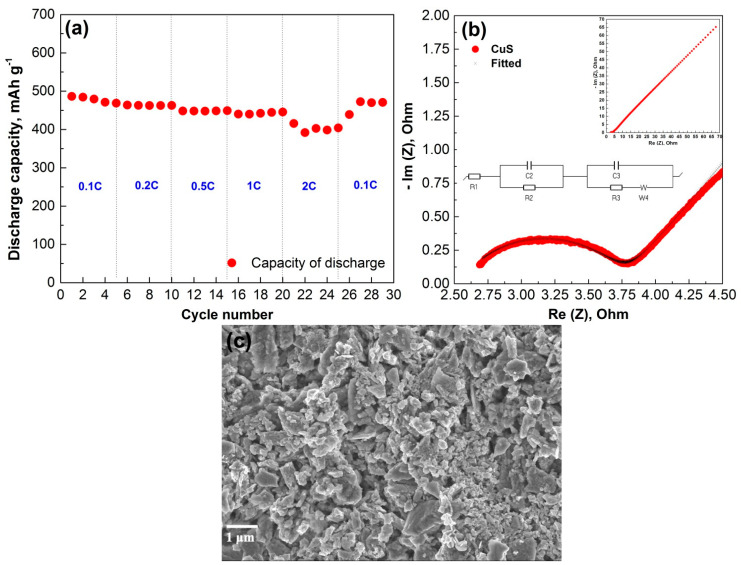
(**a**) Rate capability performance of CuS on Cu foam. (**b**) Nyquist plot after 5th cycle. (**c**) Post-mortem SEM image of CuS electrode after 5 cycles at 0.1 C.

## Data Availability

The data presented in this study are available on request from the corresponding author.

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
