# Peer review of "3D Hierarchical Nanocrystalline CuS Cathode for Lithium Batteries"

_materials, 2021, doi:10.3390/ma14071615_

Round 1

Reviewer 1 Report

It is necessary to improve the information efficiency of Figure 4 (d-g). The lattice bands with an interplanar spacing of 0.37 nm (d) should be shown in a separate figure due to poor resolution on a surface with a scale mark of 10 nm. Figures (e-g) show the same surface, but with different background colors. You can select one surface image, for example, drawing (g).

Author Response

Thank you for giving us the opportunity to submit a revised draft of our manuscript. We appreciate the time and effort that the reviewer has dedicated to providing valuable feedback on our manuscript. We have highlighted the changes within the manuscript by the red text colour.

Q. It is necessary to improve the information efficiency of Figure 4 (d-g). The lattice bands with an interplanar spacing of 0.37 nm (d) should be shown in a separate figure due to poor resolution on a surface with a scale mark of 10 nm. Figures (e-g) show the same surface, but with different background colors. You can select one surface image, for example, drawing (g).

A. We thank the reviewer for valuable comments and suggestions. Figure 4 has been updated with less confusing images of EDS mapping of CuS nanocrystalline material. 

XPS images in Figure 4d had been used separately. Please see the updated Figure 5. 

Reviewer 2 Report

Author reported the synthesis of CuS using commercially available Cu foam as a current collector and as a copper source for CuS. Further, its electrochemical performance was reported showing its flexibility for the application in Li batteries. However, it was not clear whether it can be used for Li-ion or Li-S batteries. Also, I found some information is missing and needs revision before its acceptance.

  1. Line 81 and 82, what does author meant by heating of DMSO from 80 to 115 °C? At what rate the temperature was increased to 115 °C? Similarly, how the S content was varied. Are there multiple samples prepared? Please provide clear experimental procedure that was followed for the study.
  2. Line 99, authors are suggested to provide the full chemical names used in the study.
  3. Line 118, author states that reaction takes place only at stabilized temperature 115 °C, on the other hand, sulfur solubility occur at high temperatures of above 115 °C. Is it first sulfur dissolution takes place above 115 °C and then Cu was added at 115°C? Please clarify these two statements.
  4. Please provide its performance to prove its flexibility, at different bending angles. Also, please provided the equivalent Randle circuit for the impedance spectrum.
  5. It would be better to provide reactions occurring during charge and discharge. More number of cycles are required to measure its performance.
  6. All the figures need to be cited in the text.

Author Response

Thank you for giving us the opportunity to submit a revised draft of our manuscript. We appreciate the time and effort that the reviewer has dedicated to providing valuable feedback on our manuscript. We have highlighted the changes within the manuscript by the red text colour.

Q1. Line 81 and 82, what does author meant by heating of DMSO from 80 to 115 °C? At what rate the temperature was increased to 115 °C? Similarly, how the S content was varied. Are there multiple samples prepared? Please provide clear experimental procedure that was followed for the study.

A. Thank you for the comments. Lines 81 and 82 have been updated with more details in the revised manuscript. 

Q2. Line 99, authors are suggested to provide the full chemical names used in the study.

A. Thank you for the comments. Line 99 has been updated in the revised manuscript. 

Q3. Line 118, author states that reaction takes place only at stabilized temperature 115 °C, on the other hand, sulfur solubility occur at high temperatures of above 115 °C. Is it first sulfur dissolution takes place above 115 °C and then Cu was added at 115°C? Please clarify these two statements.

A. Thank you for the comments. Line 122 has been updated in the revised manuscript. 

Q4. Please provide its performance to prove its flexibility, at different bending angles. Also, please provided the equivalent Randle circuit for the impedance spectrum.

A. Fig. 5 has been updated with more bending figures. In Fig. 8 Impedance spectra were updated with the Randles circuit in the revised manuscript. 

Q5. It would be better to provide reactions occurring during charge and discharge. More number of cycles are required to measure its performance.

A. The reactions have been added in the revised manuscript text. Please see lines 218-226 in the revised manuscript. 

Q6. All the figures need to be cited in the text.

A. Thank you for pointing this out. The revised manuscript has been checked thoroughly to avoid missing citations. 

Reviewer 3 Report

The present manuscript deals the “3D Hierarchical Nanocrystalline CuS Cathode for Lithium Batteries”. The authors prepared the CuS film directly on Cu foam using solution processing method and then characterized by XRD, SEM-EDX, TEM and XPS. Finally, they achieved the electrochemical measurements. This is a well written manuscript with meaningful results and in line with the scope of the journal. I recommend to publish this paper after addressing the following issues.

My comments are appended below.

1)How did the authors confirm the P63/mmc space group without any refinement? Please add the reference.

2)The authors should include the performance of other well-known cathode materials and compare with CuS cathode. Or at the very least, they should compare their results with the other cathode materials available in the literature (plenty of data are available for CuS film prepared by other techniques) and a detailed discussion must be added in the manuscript. Similarly, they should compare the charge transfer resistance with other well know cathodes.

3)How was the behaviour of the CuS film after cycling? Was the 3D Hierarchical structure and nanocrystalline branch maintained? Please include the SEM images after cycling and compare with as prepared film.

Author Response

Thank you for giving us the opportunity to submit a revised draft of our manuscript. We appreciate the time and effort that the reviewer has dedicated to providing valuable feedback on our manuscript. We have highlighted the changes within the manuscript by the red text colour.

Q1. How did the authors confirm the P63/mmc space group without any refinement? Please add the reference.

A. Thank you for pointing this out. The reference has been added:

[1] R.W. Potter II, H.T. Evans Jr., J. Res. U. S. Geol. Surv. 4 (1976) 205-212.

Q2. The authors should include the performance of other well-known cathode materials and compare with CuS cathode. Or at the very least, they should compare their results with the other cathode materials available in the literature (plenty of data are available for CuS film prepared by other techniques) and a detailed discussion must be added in the manuscript. Similarly, they should compare the charge transfer resistance with other well know cathodes.

A. Our synthesized CuS is very unique taking into account that thin-film without any binder or conductive agent was obtained on a 3D structured surface. To compare our results we have chosen thin CuS films to rule out the effect which might come from the binder or carbon source. The next comparison was added to the manuscript in line 202:

For example, electrodeposited 50 nm CuS thin film investigated by Chen et.al. was able to deliver initial discharge capacity equal to  545 mAh g-1  at 0.05 C rate. When the electrode is subjected to prolonged cycling the capacity drops drastically due to the solubility of lithium polysulfides [20]. Another CuS obtained by in situ preparation method reported by Wang and co-workers exhibited 400 mAh g−1 over 30 cycles [21]. 

[20] Chen, Y.; Davoisne, C.; Tarascon, J. M.; Guéry, C. Growth of Single-Crystal Copper Sulfide Thin Films via Electrodeposition in Ionic Liquid Media for Lithium Ion Batteries. J. Mater. Chem. 2012, 22 (12), 5295−5299

[21] Y.R. Wang, X.W. Zhang, P. Chen, H.T. Liao, S.Q. Cheng Electrochim. Acta, 80 (2012), pp. 264-268

Q3. How was the behaviour of the CuS film after cycling? Was the 3D Hierarchical structure and nanocrystalline branch maintained? Please include the SEM images after cycling and compare with as prepared film.

A. We thank the reviewer for this comment. The post-mortem images of the CuS electrode after cycling have been added in Figure 7c and lines 237-240. 

Round 2

Reviewer 2 Report

Author certainly made the changes according to the comments. The manuscript can be accepted for publication after minor revision.

  1. Please include the fitted parameters for the impedance spectrum. Also, it would better if diffusion coefficient of Li+ can be calculated from EIS.

Author Response

Q1: Please include the fitted parameters for the impedance spectrum. Also, it would better if diffusion coefficient of Li+ can be calculated from EIS.

We thank the reviewer for the comments. The fitted parameters were included in the text as follows in line 234:

The fitted parameters for the equivalent circuit of impedance spectra were: R1 = 2.58 Ohm, R2 = 1.236 Ohm, Q2 = 12.65e-6 F, R3 = 1.549 Ohm and Q3 = 0.074 F.

Unfortunately due to the undefined surface area of 3D Cu foam, we found it hard to calculate the Li+ ion diffusion in the electrode using the next formula: DLi=R2T2 / 2A2n2F4C2δ2 where A is the surface area of the electrode.

Reviewer 3 Report

Since the authors have clarified almost all the issues concerned, I suggest the acceptance of this paper for publication in this journal in the present form.

Author Response

We thank the reviewer for his kind comments.